# Enhanced Crude Oil Sorption by Modified Plant Materials in Oilfield Wastewater Treatment

**DOI:** 10.3390/molecules27217459

**Published:** 2022-11-02

**Authors:** Ya Shi, Liwa Ma, Shan Hou, Miao Dou, Yongfei Li, Weichao Du, Gang Chen

**Affiliations:** 1State Key Laboratory of Petroleum Pollution Control, Xi’an Shiyou University, Xi’an 710065, China; 2Shaanxi Province Key Laboratory of Environmental Pollution Control and Reservoir Protection Technology of Oilfields, Xi’an Shiyou University, Xi’an 710065, China; 3No. 4 Gas Production Plant, PetroChina Changqing Oilfield Company, Xi’an 710000, China; 4Xi’an Changqing Chemical Group Co., Ltd., PetroChina Changqing Oilfield Company, Xi’an 710000, China

**Keywords:** rice straw, quaternary ammonium cationic surfactant, surface modification, oil adsorption

## Abstract

The treatment of oilfield wastewater with high crude oil content and complex composition is a problem requiring considerable attention. In order to effectively remove crude oil contained in wastewater, in this work, rice straw, as an oil-absorbing material, was modified and used as a sorbent for crude oil. Rice straw was modified with alkali and cetyltrimethylammonium chloride (CTAC) by simple substitution reaction. The adsorption capacity of modified rice straw for oil was evaluated. The results illustrate that the adsorption rate of rice straw for crude oil was increased from 0.83 to 8.49 g/g, with the optimal conditions of 18% NaOH reacted for 90 min at 50 °C and 2% CTAC reacted for 60 min at 20 °C. The proposed modification method could be used for different materials to enhance the adsorption rate. The results of the contact angle test show that the modified straw changed from hydrophilic to hydrophobic, which may be the main reason for the improvement in the oil absorption rate. Finally, the surface structure of rice straw was characterized by X-ray diffraction (XRD), scanning electron microscopy (SEM), Fourier transform infrared spectroscopy (FTIR) and N_2_ adsorption–desorption isotherms, which further confirmed the hydrophobicity of the modified rice straw.

## 1. Introduction

A wastewater with high crude oil content, as well as exhaust gas and residue, is inevitably generated during the petroleum-refining process [1,2]. If they are not handled properly, human health could be harmed by the toxic substances in crude oil. Thus, effective treatment of crude oil and other pollutants from oilfield wastewater is a considerable challenge [3,4]. Generally, oil removal can be divided into physical, chemical and biological methods according to the principle of the treatment methods [5]. Adsorption, as a physical method, has received considerable attention for the treatment of wastewater [6,7]. Therefore, the selection of oil-absorbing materials is critical. Owing to economic considerations, cheap and efficient oil-absorbing materials are the commonly used. Thus, rice, wheat, sorghum, rape and other common crop straws are considered ideal oil-absorbing materials, with the advantages of low price, non-toxicity, and harmless and non-polluting characteristics [8,9,10,11].

Rice straw, an agricultural waste material, is an abundant cellulosic product derived from farming activity in many countries, with considerable practical significance when used an adsorbent [12,13,14]. In recent years, rice straw has commonly been used to treat oilfield wastewater and remove heavy metal ions. Sharma et al. used modified rice straw in a fixed-bed chromatography column to remove Ni(II) ions from aqueous solutions [15]. Rungrodnimitchai et al. used modified rice straw and bagasse to quickly prepare high-ion-exchange capacity biosorbent to remove heavy metals [16]. Yu et al. used straw composite materials to simulate the continuous purification of wastewater, oil–water separation and heavy metal ion removal [17]. The use of rice straw for such purification applications is not only an effective way to make full use of resources but also an effective measure to solve pollution problems at the source. However the operation of such processes is complicated and costly.

In this study, we used rice straw as an adsorbent to adsorb crude oil. The surface of rice straw is mainly composed of lignin, hemicellulose and cellulose with various reactive hydroxyl groups. Owing to the presence of hydroxyl, the surface of the lignocellulosic material is essentially hydrophilic. Therefore, the oil adsorption capacity and hydrophobicity of organic plant fibers are low, and chemical modification methods must be used to improve their crude oil adsorption capacity. A number of studies have been conducted with the aim of improving the ability of rice straw to absorb crude oil through chemical modification methods. For example, Sun et al. used acetylated straw to purify oil [15], Guo et al. used aluminum-chloride-modified rice straw biochar to intensify the dewatering of municipal sewage treatment plant sludge [18] and Tran et al. used rice straw to make aerogel for heat insulation, sound insulation and oil cleaning [19]. This series of studies shows that the adsorption capacity of modified straw can be improved. To the best of our knowledge, the use of quaternary ammonium salt cationic surfactants with varying alkyl chain lengths to modify straws as natural adsorbents for crude oil in wastewater has not been reported to date.

Therefore, inspired by the abovementioned research, in this work, our goal was to improve the oil adsorption ability of rice straw and obtain a highly efficient modified oil-absorbing material by a simple, easy-to-operate and low-investment method. In this study, rice straw was modified by quaternary ammonium salt-type cationic surfactants with varying alkyl chains in an effort to improve the hydrophilic properties of the straw. Hydrophobic characteristics were monitored with a contact angle measuring instrument. The surface morphology was characterized by Fourier transform infrared spectroscopy (FTIR), X-ray diffraction (XRD), scanning electron microscopy (SEM) and N_2_ adsorption–desorption isotherms.

## 2. Results and Discussion

### 2.1. Mechanism of Modification

The mechanism of modification is exhibited in Figure 1. First, the straw was treated with alkali. In the reaction mechanism, the alkali cellulose and lignin formed by cellulose and lignin on the surface of rice straw react with concentrated alkali solution (Figure 1(1)), causing the cellulose and lignin to produce unstable cations on the surface of the fiber so as to react with the nucleophile.

The second part is the modification of cationic surfactants. The reaction mechanism is the adsorption of anion and cation charges. However, the raw materials must be washed to neutrality with deionized water after the first step of the alkali treatment because the surfactant easily loses activity and precipitates in the alkaline medium. In this mechanism, the positively charged quaternary ammonium nitrogen atom on the cationic surfactant combines with the negatively charged oxygen atom on the hydroxyl through electrostatic interaction to modify the surface of the straw so as to improve the oil absorption value (Figure 1(2)).

### 2.2. Effect of Modification Conditions on Oil Absorption

#### 2.2.1. Effect of Alkali Modification Conditions

First, the effect of alkali type was examined under a concentration of 18% with a 90 min reaction at 50 °C. As shown in Figure 2a, the NaOH has the highest oil absorption value, reaching 5.10 g/g because the NaOH can undergo a saponification reaction with the lignin on the surface of the straw, increasing the active sites on the surface of the straw and improving the oil adsorption performance of rice straw [20]. Therefore, NaOH was selected as the treatment agent for alkali treatment.

The effect of NaOH was examined by varying the concentration from 0 to 22% after a 90 min reaction at 50 °C. Figure 2b shows that the oil absorption value increases with increased NaOH concentration. The oil absorption value reaches a maximum when the concentration of NaOH reaches 18%. The overall oil absorption value increases from 1.90 to 5.10 g/g when the concentration of NaOH increases from 0 to 18%. The reaction can be easily carried out when more anions are produced the reaction between the cellulose and lignin on the surface of the rice straw with increased NaOH concentration. Therefore, 18% NaOH was selected as the optimal concentration.

As shown in Figure 2c, the effect of the alkali treatment temperature on the oil adsorption ability was studied under an 18% concentration of NaOH with a 90 min reaction. Figure 2c shows that the oil absorption value increases from 2.90 to 4.14 g/g when the NaOH treatment temperature is lower than 50 °C. However, the oil absorption value does not increase significantly and remains stable when the temperature is above 50 °C because the oil adsorption reaches saturation at 50 °C. Therefore, 50 °C was selected as the optimal NaOH treatment temperature.

To optimize the NaOH treatment time, the effect of NaOH treatment time was investigated under an 18% at 50 °C. Figure 2d shows that the oil absorption value increases steadily from 3.46 to 5.29 g/g when the duration of alkali treatment is between 0 and 90 min. Moreover, the oil absorption value decreases significantly from 5.29 to 3.80 g/g when the time exceeds 90 min, as the adsorption equilibrium is reached quickly, and the oil absorption value continues to increase with the increment of reaction time from 0 to 90 min. However, the adsorption balance is destroyed and the oil absorption value decreases when the time exceeds 90 min. An excessively long alkali treatment time could also cause the phenolic hydroxyl group in the lignin in the rice straw disappear, which is unfavorable for subsequent surfactant modification [21]. Therefore, 90 min was selected as the optimal length of NaOH treatment.

#### 2.2.2. Effect of Cationic Surfactant Modification Conditions

We first investigated the effect of quaternary ammonium cationic surfactant type on oil adsorption under a 2% concentration with a 60 min reaction at 20 °C. Figure 3a shows the effects of different cationic surfactants on rice straw; the oil absorption value of surface modification is highest with CTAC. The oil absorption value of DTAC is 7.54 g/g, that of CTAC is 8.49 g/g, that of OTAC is 7.61 g/g and that of BTAC is 6.83 g/g. As shown in Table 1, that the contact angle of CTAC is the largest, owing to the decrease in surface activity and water solubility with increased hydrophobic carbon chain length. Therefore, CTAC was selected as the optimal surface modification treatment agent.

Then, the concentration of CTAC was examined at 20 °C with a 60 min reaction by varying the concentration from 1 to 6%. Figure 3b shows that the oil absorption value increases rapidly from 3.00 to the highest value of 5.23 g/g when the concentration of CTAC increases from 1 to 2%, mainly because oil adsorption reaches saturation when the concentration is 2%. However, as the concentration continues to increase, the oil absorption value is considerably reduced. Therefore, the optimal concentration of CTAC was selected as 2%.

The modification temperature of CTAC was investigated under a 2% CTAC concentration with a 60 min reaction. As shown in Figure 3c, the maximum oil absorption value of 6.53 g/g was reached with a surface modification temperature of 20 °C. The oil absorption value was considerably reduced from 6.53 to 4.00 g/g with increasing temperature because cationic surfactants are prone to inactivation at high temperatures, so the reaction is more likely to be successful at low temperatures. Although we can speculate that the lower temperature, the higher adsorption rate, moderate temperature or room temperature is most easily applicable; therefore, 20 °C was selected as the surface modification temperature.

Finally, the effect of CTAC modification time was investigated under a concentration of 2% at 20 °C. Figure 3d shows that the oil absorption value significantly increases from 3.64 to 8.49 g/g when the surface modification time is between 0 and 60 min. However, the oil absorption value rapidly declines from 8.49 to 5.30 g/g when the time exceeds 60 min because the ion movement of the cationic surfactant is most violent with a 60 min reaction time, and the crude oil will be resolved after this time. Therefore, 60 min was selected as the optimal length of surface modification

### 2.3. Comparison of Agricultural Straw Materials

In order to identify a suitable, economical and efficient absorbing material, in this study, five natural plant materials were selected for the same alkali treatment and CTAC modification; then, their oil absorption values were measured and compared. Table 2 shows the oil absorption values of five plant materials. The oil adsorption of alkali and CTAC modified natural plant materials improved considerably, with the greatest improvement in straw, from 0.83 to 8.49 g/g. This is because alkali treatment can increase the exchangeable cations on the straw surface, resulting in a loose and lipophilic material [22]. Moreover, the oil adsorption of modified rice straw far exceeds that of other plant materials under the same conditions. Therefore, rice straw was selected as an optimal oil-absorbing material.

### 2.4. Characterization of Straw

In order to determine the change in the contact angle between the materials with deionized water before and after treatment, the materials were analyzed by a contact angle measurement instrument. Figure 4 and Table 3 show the contact angle between materials and deionized water, with Figure 4a,b representing the contact angle between unmodified and modified materials, respectively, with deionized water. Figure 4b shows that the contact angle between various materials and deionized water significantly increases from 0° to about 70° owing to CTAC modification, demonstrating that the modified materials exhibit hydrophobic characteristic [23], with rice straw exhibiting increased hydrophobicity. The modified rice straw displays the highest oil adsorption ability, which can be attributed to its hydrophobicity and the porosity of its surface [24].

The rice straw was characterized using an FTIR spectrometer before and after modification. Figure 5 shows that there is no significant difference between the unmodified and modified peaks. However, the intensity of the C-H band at 2800–2900 cm^−1^ [7,20,23] increased, whereas other peaks weakened, owing to the substitution reaction of NaOH and the CTAC compound modification treatment.

The XRD patterns of unmodified and modified rice straw shown in Figure 6 exhibit characteristic peaks around 2θ of 22.5°, typically representing cellulose Iβ lattice [25,26]. Moreover, the crystallinity after modification is weaker than that before treatment because the intermolecular and intramolecular bonds of cellulose were destroyed by the composite modification treatment with NaOH and CTAC, leading to a decrease in crystallinity [27].

The N_2_ adsorption–desorption isotherms of unmodified and modified rice straw were achieved according to the Brunauer–Emmett–Teller (BET) method. The surface areas, pore volumes and pore sizes are listed in Table 4. As shown in Figure 7a, the N_2_ adsorption–desorption isotherm of rice straw is a IV hysteresis isotherm [28]. As shown in Figure 7b (insert), most pores have a size of 2.3 nm. The data presented in Table 4 further demonstrate that the pore properties of modified rice straw were improved, and the BET surface area increased from 6.68 to 43.46 m^2^/g, which further supports the high oil-adsorption capacity of modified rice straw.

Finally, in order to observe the morphology and microstructure of rice straw before and after modification, the material was analyzed by SEM. Figure 8 shows the tubular structures of unmodified (a),(b) and modified (c),(d) rice straw. Figure 8c shows that modified rice straw is easier to separate into a single tubular structure, and Figure 8d shows the porous and loose web structure of the fibers. A comparison of Figure 8 shows that the surface area of the rice straw increased, enhancing its hydrophobic properties [29] and supporting the contact angle measurements and N_2_ adsorption–desorption data reported earlier in this article.

## 3. Materials and Methods

### 3.1. Materials

The oil used in the experiment was collected from Chang 2 reservoir of Yanchang Oilfield, the properties of which are shown in Table 5. Dodecyl trimethyl ammonium chloride (DTAC), cetyltrimethylammonium chloride (CTAC), octadecyl trimethyl ammonium chloride (OTAC), behenyl trimethyl ammonium chloride (BTAC) and other reagents used were of analytical grade and were used as purchased without further purification. Rice straw was collected from farmland around Jingmen City, Hubei Province.

### 3.2. Methods

#### 3.2.1. Pretreatment of Rice Straw

The rice straw was cut into pieces 1–2 cm in length and soaked in deionized water for 2 h; then, the samples were cleaned once with distilled water and placed in an oven at 65 °C to achieve constant weight [7]. A schematic representation of the pretreatment process is shown in Figure 9.

#### 3.2.2. Modification of Rice Straw

A schematic representation of the modification process of rice straw is shown in Figure 10. First, the rice straw was treated with alkali. A volume of 50 mL of lye was added to the beaker with oven-dried straw; then, the beaker was placed in a water bath at 50 °C for 90 min. The reaction was terminated after the required time by removing the lye and washing the samples with deionized water until the straw was neutral. Samples were then dried in an oven at 65 °C for 24 h.

Then, the oven-dried rice straw in the beaker was modified with 50 mL quaternary ammonium cationic surfactant solution, and the beaker was placed in in a water bath at a 20 °C for 60 min. Samples were then dried in an oven at 65 °C for 24 h for the use for adsorption.

### 3.3. Determination of Oil Absorption Value

The dried straw was placed into a homemade small iron cage of about 30 mesh and then into a 250 mL beaker containing 150 mL of oil at 40 °C. The iron cage was removed after 90 min so as to drip oil within 40 min. Finally, the straw was weighed, and three readings and weight measurements were taken for each sample to verify the repeatability and to obtain an average value. The oil absorption value was calculated as follows:Q = (m_2_ − m_1_)/(m_1_ − m_0_)(1)
where Q is the oil absorption value (g/g), m_0_ is the mass of the blank net (g), m_1_ is the total mass of the net and the oil-absorbing material before oil adsorption (g) and m_2_ is the total mass (g) of the net and the oil-absorbing material after oil adsorption. The oil adsorption of the blank net was subtracted to calculate the final oil adsorption.

### 3.4. Characterization of Rice Straw

The contact angle was determined by using a contact angle measurement instrument (JC2000DS, Shanghai, China) [30]. The infrared spectra of unmodified and modified rice straw were analyzed using an FTIR spectrometer. All IR measurements were conducted using the KBr pellet technique (1 mg of sample homogenized with 200 mg KBr) with a Fourier infrared spectrometer at room temperature in the range of 400 to 4000 cm^−1^, with 32 scans recorded per sample at a resolution of 0.4 cm^−1^.

The morphology was obtained with powdered samples by means of a field emission scanning electron microscope (SEM, JSM-6390A, JEOL, Tokyo, Japan). The crystallinity of the rice straw was analyzed by powder X-ray diffraction (XRD) (JDX-3530, JEOL, Japan) on an XRD-6000 diffractometer with Cu Kα radiation at 40 kV voltage and 15 mA current. N2 adsorption–desorption isotherms were performed on a Micrometrics ASAP 2020 HD88 instrument (Norcross, GA, USA) at 77 K, for which the samples were degassed at 573.15 K for 4 h before the measurement. The surface area and pore structure were calculated utilizing the Brunauer–Emmett–Teller (BET) method and the Barrett–Joyner–Halenda (BJH) model, respectively.

## 4. Conclusions

In this study, rice straw was used as a raw material, and NaOH and CTAC were used as a surface modification treatment agent. The adsorption capacity of crude oil was evaluated through specific oil adsorption experiments. The following optimal preparation conditions were determined: 18% NaOH reacted for 90 min at 50 °C and 2% CTAC reacted for 30 min at 20 °C. The reason for the increased crude oil sorption ability of the modified material was explained at the theoretical level of the size of contact angle and the principle of wettability. By comparing different types of plant materials modified for oil adsorption magnification and contact angle before and after modification, modified rice straw with the highest oil adsorption effect was determined as a high-efficiency oil-absorbing material, and the crude oil adsorption ratio increased from 0.83 to 8.49 g/g. XRD, FTIR, SEM, contact angle measurement and N_2_ adsorption–desorption characterization further proved that hydrophobicity of the modified rice straw. This results reported herein will be of use for related work in oilfield waste treatment.

## Figures and Tables

**Figure 1 molecules-27-07459-f001:**
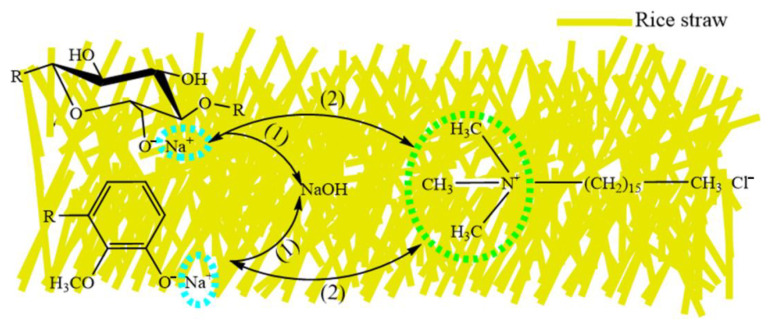
Mechanism of modification reaction.

**Figure 2 molecules-27-07459-f002:**
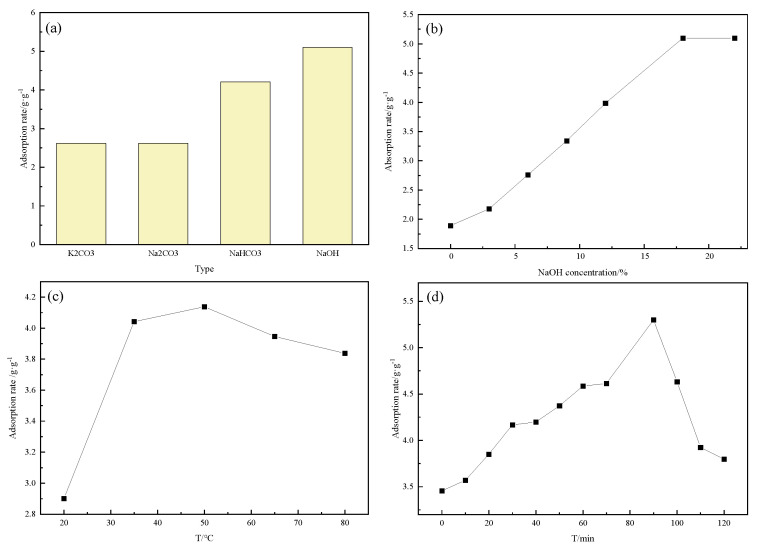
Effect of alkali modification conditions: alkali type (**a**), NaOH concentration (**b**), treatment temperature (**c**) and treatment time (**d**).

**Figure 3 molecules-27-07459-f003:**
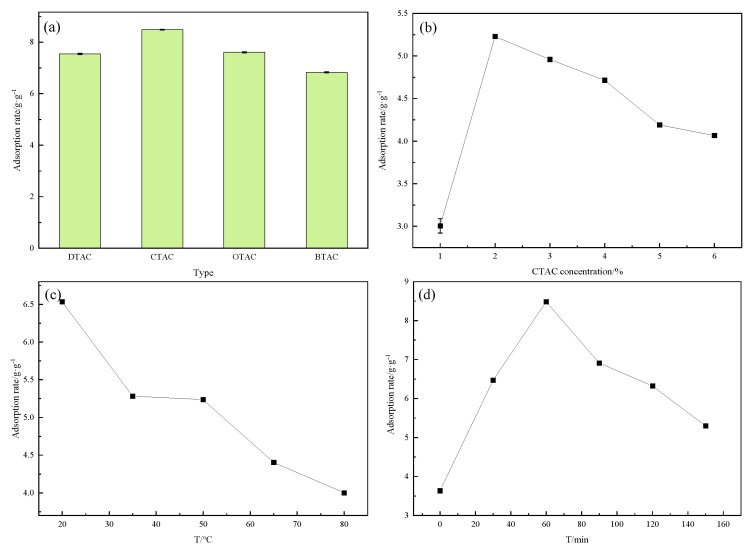
Effect of cationic surfactant modification conditions: cationic surfactant type (**a**), CTAC concentration (**b**), surface modification temperature (**c**) and surface modification time (**d**).

**Figure 4 molecules-27-07459-f004:**
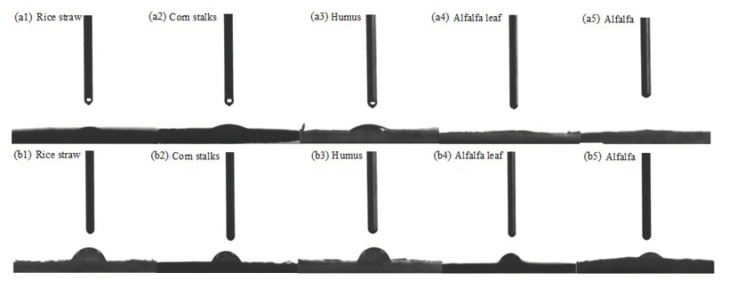
Contact angle between unmodified and modified plants.

**Figure 5 molecules-27-07459-f005:**
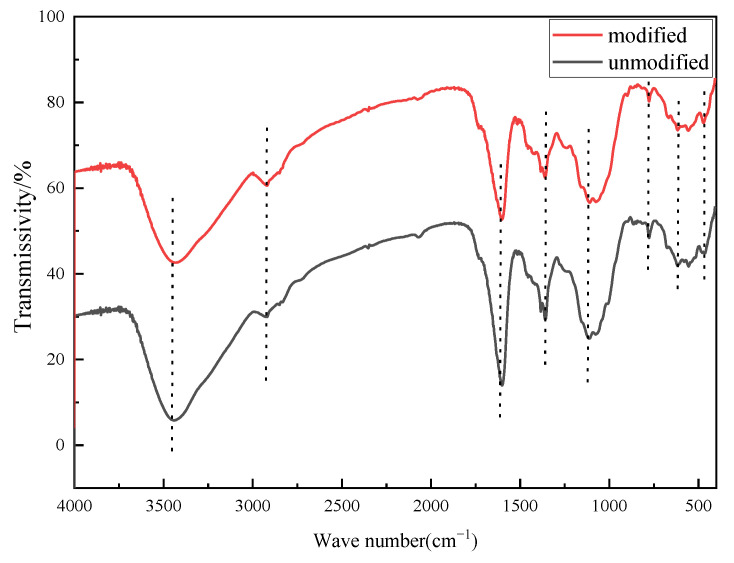
FT−IR spectra of unmodified and modified rice straw.

**Figure 6 molecules-27-07459-f006:**
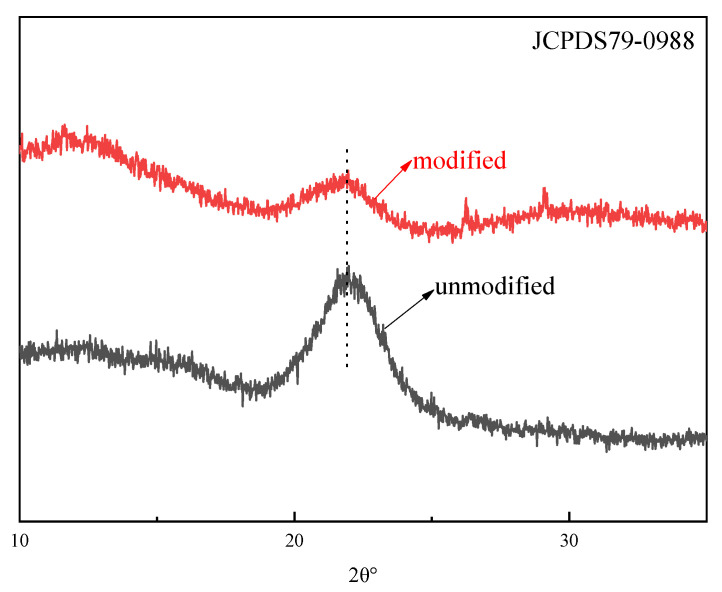
XRD pattern of unmodified and modified rice straw.

**Figure 7 molecules-27-07459-f007:**
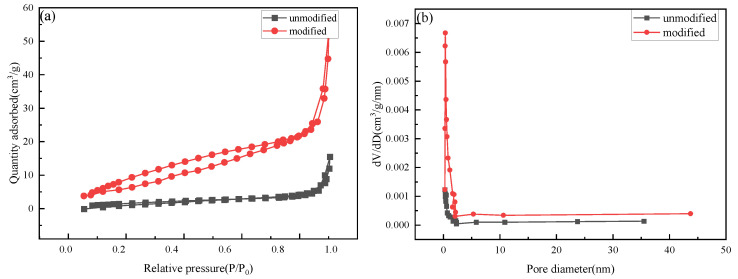
N_2_ adsorption–desorption isotherms and pore size distribution of unmodified and modified rice straw: N_2_ adsorption–desorption isotherms (**a**), pore size distribution (**b**).

**Figure 8 molecules-27-07459-f008:**
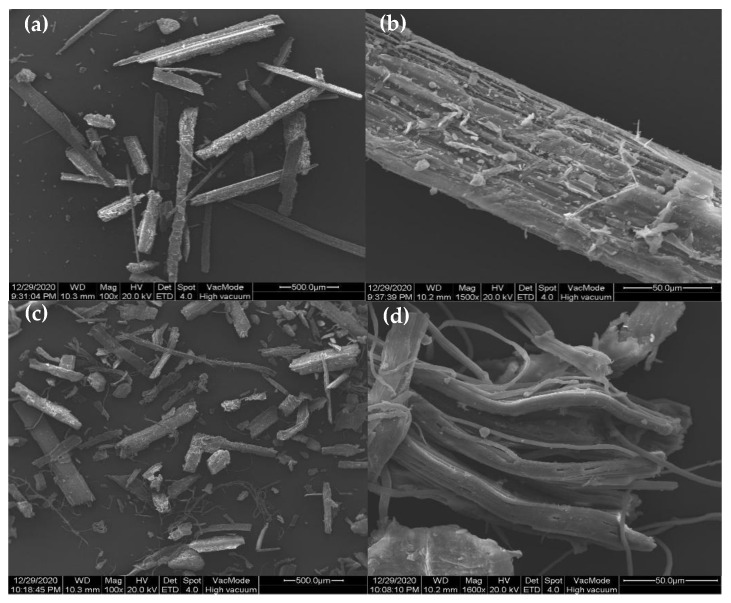
SEM images of unmodified (**a**,**b**) and modified (**c**,**d)** rice straw.

**Figure 9 molecules-27-07459-f009:**
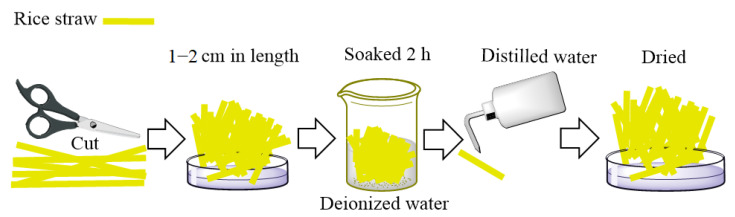
Preparation of rice straw samples.

**Figure 10 molecules-27-07459-f010:**
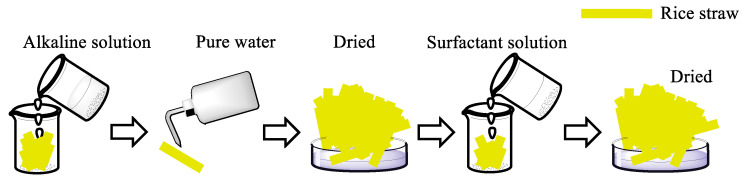
Modification of rice straw samples.

**Table 1 molecules-27-07459-t001:** Contact angle value of the surfactant.

Surfactant	DTAC	CTAC	OTAC	BTAC
Contact angle	88°	95°	84°	79°

**Table 2 molecules-27-07459-t002:** Summary of the oil adsorption of various oil-absorbing materials.

No.	Material	Oil Absorption Value (g/g)
1	Rice straw (unmodified)	0.83
2	Alfalfa (modified)	3.06
3	Alfalfa leaf (modified)	3.91
4	Humus (modified)	6.21
5	Corn stalk (modified)	6.52
6	Rice straw (modified)	8.49

**Table 3 molecules-27-07459-t003:** Contact angle of unmodified and modified straw.

	Rice Straw	Corn Stalk	Humus	Alfalfa	Alfalfa Leaf
unmodified	10.20°	19.20°	21.25°	4.38°	4.62°
modified	70.10°	71.02°	72.28°	68.20°	43.25°

**Table 4 molecules-27-07459-t004:** Pore properties of unmodified and modified rice straw.

Pore Properties	BET Surface Area (m^2^/g)	Pore Volume (cm^3^/g)	Pore Size (nm)
unmodified	6.67	0.024	14.92
modified	43.46	0.101	10.24

**Table 5 molecules-27-07459-t005:** Physical parameters of oil from Yanchang oil field.

μ30 /(mPa·s)	Pour Point t/°C	ρ20 /(g·cm^−3^)	Resins W/%	Asphaltenes W/%	Aromatic Hydrocarbons W/%	Saturated Hydrocarbons W/%
36.9	18.5	0.86	12.1	6.8	25.2	55.9

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
