# Peer review of "Enhanced Crude Oil Sorption by Modified Plant Materials in Oilfield Wastewater Treatment"

_molecules, 2022, doi:10.3390/molecules27217459_

Round 1
Reviewer 1 Report
COMMENTS TO THE AUTHORS
This paper deals with the systematic study of modified rice straws with alkali and cetyltrimethylammonium chloride for crude oil removal from wastewater. This modification improved the crude oil adsorption rate. Various analytical, spectroscopic, and microscopic techniques were adopted to analyze the modified samples. The presented results are consistent, and the submitted manuscript contains only a few misunderstandings that need to be corrected. The paper is therefore recommended for publication after a minor revision.
1. In the Abstract section, is “powder” a typo error in the sentence “Finally, the surface structure of rice straw was characterized by powder…”? Did cetyltrimethylammonium chloride modify rice straws, or was it used as a surfactant in wastewater? It is unclear.
2. In the Discussion section, is any specific function for fitting scatters (connected by lines) in Figures 2 and 3? Maybe better to remove these connecting lines. What is the type of bonding between rice straw and cetyltrimethylammonium chloride, and what is the stability of this modification? Why did the adsorption rate decrease with increasing temperature for modification using surfactant? Values of the contact angle of water are missing. A description of the y-axis is missing in Figure 5. Did BET provide information only about the porosity of a particular fiber or complex of fibers?
3. In the Experimental section, there is a lack of information about used techniques and materials, for example, how the contact angle of water was measured (sessile drop method, Washburn), or missing suppliers. What was the sample preparation for this type of measurement to ensure a smooth substrate? Should “quality” in “m0 is the quality of the empty net” be corrected?
Author Response
- In the Abstract section, is “powder” a typo error in the sentence “Finally, the surface structure of rice straw was characterized by powder…”? Did cetyltrimethylammonium chloride modify rice straws, or was it used as a surfactant in wastewater? It is unclear.
The issues raised have been revised in the text.
- In the Discussion section, is any specific function for fitting scatters (connected by lines) in Figures 2 and 3? Maybe better to remove these connecting lines. What is the type of bonding between rice straw and cetyltrimethylammonium chloride, and what is the stability of this modification? Why did the adsorption rate decrease with increasing temperature for modification using surfactant? Values of the contact angle of water are missing. A description of the y-axis is missing in Figure 5. Did BET provide information only about the porosity of a particular fiber or complex of fibers?
(1) The problems raised have been modified in Fig. 2 and Fig. 3.
(2) The bonding type between rice straw and cetyltrimethylammonium chloride is intermolecular electrostatic interaction, and this modification has strong stability.
(3) This is because cationic surfactant is easy to decompose or desorb at high temperature, so when surfactant is used for modification, the adsorption rate will decrease with the increase of temperature.
(4) Table 2 has been added to supplement the contact angle of water. A description has been added to y-axis in Figure 5.
(5) BET provided the porosity of rice straw before and after modification in this study, which showed that the pore characteristics of modified rice straw were improved, and further showed that the modified rice straw had high oil adsorption capacity.
- In the Experimental section, there is a lack of information about used techniques and materials, for example, how the contact angle of water was measured (sessile drop method, Washburn), or missing suppliers. What was the sample preparation for this type of measurement to ensure a smooth substrate? Should “quality” in “m0 is the quality of the empty net” be corrected?
The measurement of contact angle has been supplemented with references.
The “quality” in “m0 is the quality of the empty net”, which has been corrected to “mass”.
Reviewer 2 Report
Dear Authors
The presented work in your manuscript is interesting for the readers and contains significant data of importance in the field of oil spill removal which directly has an impact on the environment quality for the living organism and humans.
The idea of the work is not novel, however, the main point is the use of waste of low cast to develop material of added value that can be used in the improvement of the life quality.
The research design is lacking in studying some important factors which have a direct impact on the oil adsorption capacity of the developed rice straw matrix. For example, the length of the rice straw. This leads us to raise a question about why the authors did not use a powder form of the modified rice straw to get a much higher surface area? It is known that the adsorbent surface area is a determining factor.
General comments
why do the authors correlate the variation of the oil adsorption capacity to the variation of the treatment conditions? It is an indirect relation. The appropriate correlation should be presented between the hydrophobicity of the modified rice straw, contact angle measurements and water uptake, and the oil adsorption capacity.
The authors did give a justification for using different surfactants. What is the difference between them? and what is the impact of this difference on the efficiency of oil adsorption capacity?
specific comments
2.2.1. section
The authors need to give an explanation of the obtained behavior during the study of the alkali treatment time specifically when the treatment time exceeded 90 minutes.
2.2.2. section
The authors did not give any explanation of the superiority of the sample treated with CTAC surfactant over other used ones in the study. The authors must give the hydrophobicity measures and correlate them to the samples' adsorption capacity to explain why the CTAC-treated sample is the best one.
2.3. section
The authors have to give a reason for the rice straw-treated sample with CTAC superiority over other natural plant materials in the oil adsorption capacity.
3.3. section
The oil adsorption rate is a relation between the amount of adsorbed oil and the adsorption time. The authors in this section mentioned the oil adsorption capacity (g/g). Please correct.
In conclusion, a major revision is needed before reconsidering your manuscript for publicatopn.
Author Response
specific comments
2.2.1. section
The authors need to give an explanation of the obtained behavior during the study of the alkali treatment time specifically when the treatment time exceeded 90 minutes.
The following explanations have been made in the text with reference to the literature:
However, the adsorption balance would be destroyed and the oil adsorption rate decreases when the time exceeds 90min. Too long alkali treatment time will also make the phenolic hydroxyl group in lignin in rice straw disappear, which will be unfavorable for subsequent surfactant modification[30].
2.2.2. section
The authors did not give any explanation of the superiority of the sample treated with CTAC surfactant over other used ones in the study. The authors must give the hydrophobicity measures and correlate them to the samples' adsorption capacity to explain why the CTAC-treated sample is the best one.
The measured value of contact angle has been added in the text, and the following explanation has been made for why the sample treated by CTAC is the best one:
It can be seen from the contact angle of the four surfactants in Table 1 that the contact angle CTAC is the largest, which is due to the decrease of surface activity and water solubility with the increase of hydrophobic carbon chain length.
2.3. section
The authors have to give a reason for the rice straw-treated sample with CTAC superiority over other natural plant materials in the oil adsorption capacity.
The following explanations have been made in the text with reference to the literature:
This is because alkali treatment can increase the exchangeable cation on the straw surface and make it loose and lipophilic[29].
3.3. section
The oil adsorption rate is a relation between the amount of adsorbed oil and the adsorption time. The authors in this section mentioned the oil adsorption capacity (g/g). Please correct.
“oil adsorption rate” should be oil absorption value. We have revised it.
Round 2
Reviewer 2 Report
Dear Authors
The revised version of your submitted manuscript can be accepted for publication after taking into consideration the raised comments and adding further data to improve the quality and clarity of the manuscript.